# Development of Non-Destructive Testing Device for Plant Leaf Expansion Monitoring

Xianchang Meng [1,2,3], Yili Zheng [1,2,3] and Weiping Liu [1,2,3,*]

1 School of Technology, Beijing Forestry University, Beijing 100083, China
2 Beijing Laboratory of Urban and Rural Ecological Environment, Beijing Municipal Education Commission, Beijing 100083, China
3 Key Lab of State Forestry Administration for Forestry Equipment and Automation, Beijing 100083, China
* Correspondence: hfpl916@126.com

**Abstract:** This paper designs a plant leaf expansion pressure non-destructive detection device, aiming to promote plant leaf expansion pressure research and achieve precision irrigation. The design is based on leaf expansion pressure probe technology, which can effectively monitor the plant leaf expansion pressure by detecting the feedback of the leaf under constant pressure. In this paper, the stability of the sensor and the calibration model is tested. The calibration experiments showed that the coefficient of determination $R^2$ of the sensor was over 0.99, the static test results showed that the range of the sensor was 0–300 kPa, and the fluctuation of the sensor was less than 0.2 kPa during the long-term stability test. The indoor comparison tests showed that there was a significant difference in the variation of leaf expansion pressure data between plants under drought conditions and normal conditions. The irrigation experiments showed that the leaf expansion pressure was very sensitive to irrigation. The correlation between the expansion pressure data and the environmental factors was analyzed. The correlation coefficient between expansion pressure and light intensity was found to be 0.817. The results of the outdoor experiments showed that there was a significant difference in the expansion pressure of plants under different weather conditions. The data show that the plant leaf expansion pressure non-destructive detection device designed in this paper can be used both as an effective means of detecting plant leaf expansion pressure and promoting the research of plant physiological feedback mechanisms and precision irrigation.

**Keywords:** sensor; non-destructive testing; leaf expansion; water potential





## 1. Introduction

Plant leaf water potential is an important indicator of overall plant water status. Compared with soil water content and stem fluid flow, plant leaf water potential can provide more accurate and rapid feedback on plant water changes because it is in the middle of the Soil–Plant–Atmosphere Continuum system's water potential and is a key link for accurate irrigation [1–6].

In early studies, leaf water potential was measured mainly by the pressure chamber method, small liquid flow method, thermocouple method, and xylem pressure probe method [7,8]. However, these methods are often unable to achieve in situ measurements, have the disadvantage of being destructive to the plant body, and have a lag in the measurement results.

With the development of spectroscopy technology, techniques such as near-infrared spectroscopy and terahertz spectroscopy have been used to determine the water potential of plant leaves [9–14]. These methods have the advantages of fast measurement, high image accuracy, and easy visual analysis, but it is difficult to achieve real-time detection for the actual production of agriculture and forestry, and the equipment is expensive. Therefore, a plant leaf water potential device that is easy to install, inexpensive, and can be monitored

online in real time has become an urgent need. Zimmermann et al. proposed a detection method based on plant leaf bulking pressure: the leaf bulking probe technique (LPCP) [15]. The only commercially available product is the German ZIM-probe (YARA ZIM Plant Technology GmbH). This equipment has been used by domestic and foreign scholars to conduct research on the expansion of many plants, such as burlap, olive, wheat, rape, banana, nectarine, and tomato plants [16–25]. However, due to its high cost and limited scope of application, the installation is more complex, and the automation capability is poor, which limits the application's promotion.

This paper is based on the leaf expansion probe technique, leaf cell characteristics, and the Wheatstone bridge principle. A leaf expansion pressure non-destructive testing device is designed. The device is based on the STM32 microprocessor as the core; achieves long-term, stable, and accurate monitoring of leaf expansion pressure; and uses LoRa (a medium-range wireless communication technology) to achieve the wireless transmission of data. The device also measures ambient environmental factors, such as temperature and humidity. Useful for comparative analysis with blade expansion pressure data, the device is inexpensive, designed to cost only 1/5 of current commercial products, and useful in promoting leaf expansion-pressure-related research.

## 2. Materials and Methods

### 2.1. Measurement Principle

When the plant water potential changes, the first to change is the expansion pressure of the leaf. From a microscopic point of view, the plant leaf expansion pressure $P_c$ is obtained by the cell osmotic pressure, cell wall elasticity, cell membrane permeability, extracellular liquid static pressure, and other comprehensive impacts when the sensitivity to the water status is stronger; from a macroscopic point of view, the calculation formula of $P_c$ is

$$P_c = \Psi_s + \pi - g_s \cdot VPD_L / K \tag{1}$$

where $\Psi_s$ is the soil water potential, $VPD_L$ is the water vapor pressure difference between the leaf surface and the atmosphere, $K$ indicates the hydraulic conductivity between the soil and the leaf, and $g_s$ is the leaf stomatal conductivity.

From the above equation, it can be seen that the variation in leaf expansion pressure is related to root water uptake and stomatal conductance, so it can comprehensively characterize the water status of plants.

The sensor structure is shown in Figure 1. The vane expansion pressure sensor uses the pressure output of the sensed vane at a constant pressure to determine the pressure value [26]. The leaf under testing is placed between two magnetic cylinders, one of which contains a micro-pressure sensor inside and is surrounded by a silicone oil filling. The pressure $P_{clamp}$ is applied to the leaf by the magnetic force between the cylinders, and the leaf expansion pressure $P_c$ is transmitted to the pressure sensor via silicone oil to measure the relative leaf expansion pressure $P_p$.

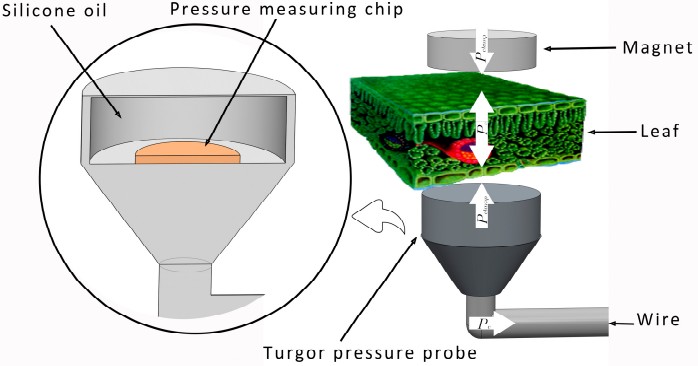

**Figure 1.** Sensor system structure diagram.

The physical sensor is shown in Figure 2. It consists of a magnet, a turgor pressure probe, and a wire. When in use, the blade is clamped between the magnet and the turgor pressure probe. The measured data are transmitted to the STM32 microprocessor through wires.

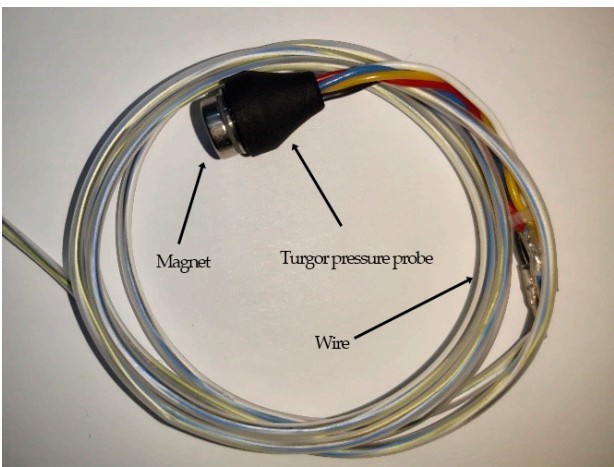

**Figure 2.** Physical view of sensor probe.

The presence of a magnet inside the pressure probe generates a magnetic force. When clamping the blade, due to the cellular characteristics of the blade, and the loss in the process of pressure transmission, it will lead to a slightly lower detection value than the actual value. At this time, the pressure value of the input blade is calculated as follows:

$$P_{in} = F_a \cdot P_{clamp} \tag{2}$$

where $P_{in}$ is the pressure received by the leaf cell, $F_a$ is the pressure-decay factor, and $P_{clamp}$ is the magnetic suction force provided by the probe.

When the pressure signal $P_{clamp}$ of the sensor is unchanged, the pressure received by its leaf cell depends only on the pressure-decay factor. For measurements on different plants, the pressure probe's internal transmission medium remains the same, and its attenuation effect on pressure remains the same so that the attenuation factor changes only depend on the leaf characteristics of the different plants [23]. For the same plant, the pressure received by the leaf cells is also fixed so that the relative leaf expansion pressure $P_p$ measured by the sensor is only related to the cellular transfer function $T_f(V)$ of the plant leaf cells. The relationship between the relative leaf expansion pressure $P_p$ detected by the sensor and the pressure signal received by the leaf cell is

$$P_p = T_f(V) \cdot P_{in} \tag{3}$$

There is also a functional relationship between the expansion pressure $P_c$ and the volume $V$. In the study of plant cell mechanics, the relationship is

$$\varepsilon_p = V \cdot \frac{dP_c}{dV} \tag{4}$$

where $\varepsilon_p$ is the average volumetric elastic modulus, whose main variation depends on the expansion pressure $P_c$ [27–29], which is set as a linear relationship:

$$\varepsilon_p = a \cdot P_c + b \tag{5}$$

where $a$, $b$ is the characteristic constant of the leaf, as in the study of Chlorella cells by Zimmermann et al. [30]. It can be concluded that both a and b are less than 1, and the constant is relatively large when the cells show rapid pressure changes, and relatively small

for slow pressure changes. Rectifying the equation by combining (2)–(5), we obtain the following functional relationship:

$$P_p = \left( \frac{b}{a \cdot P_c + b} \right)^{\frac{1}{a}} \cdot P_{in} = \left( \frac{b}{a \cdot P_c + b} \right)^{\frac{1}{a}} \cdot F_a \cdot P_{clamp} \tag{6}$$

In the sensor probe's internal use of the Wheatstone bridge circuit to measure the change in pressure, the leaf feedback pressure will make the bridge strain resistance deform, thereby changing the output voltage, to obtain the amount of expansion pressure change. At the same time, in the probe's internal integration of the high-precision ADC chip, the bridge voltage changes directly through the internal ADC chip for analog-to-digital conversion, and the output digital quantity goes to the MCU to avoid the analog signal transmission process having some of the influencing factors caused by the error.

## 2.2. Hardware System Design

### 2.2.1. Hardware Circuit General Structure

The overall structure of the hardware circuit of the leaf expansion sensor is shown in Figure 3, mainly including the STM32 processor and core circuit, power supply module, voltage detection module, SD card storage module, LoRa communication module, and sensor monitoring module. Among them, the STM32 processor is the core unit of the whole-leaf expansion pressure sensor. It is responsible for driving the sensor unit and transferring the collected sensor information for processing.

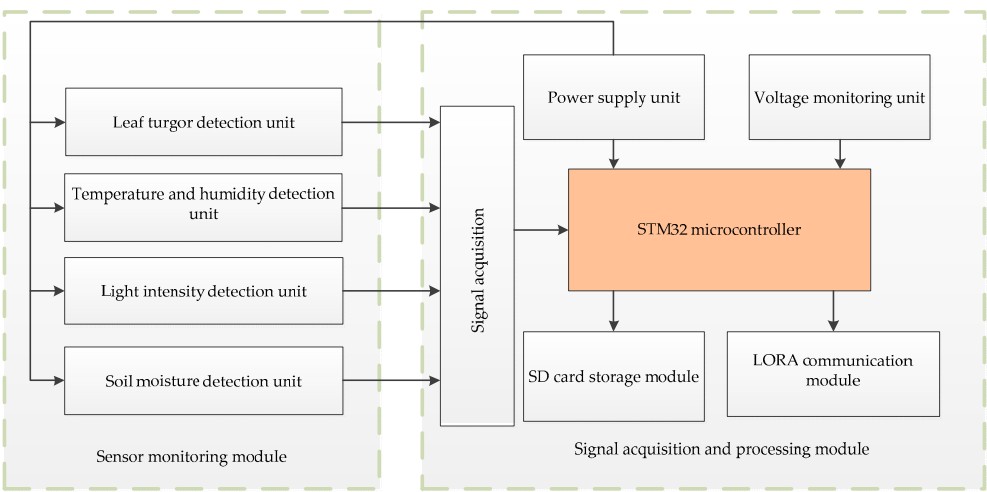

**Figure 3.** System architecture diagram.

The power supply module provides effective power input for the STM32 microcontroller, and the STM32 processor performs checks on each module to ensure that each module is working properly. Then, the STM32 processor issues query commands to the sensor, which transmits the collected data to the STM32 processor through the IIC communication protocol or SPI communication protocol and performs further processing to complete data conversion, storage, and communication.

The simplified circuit diagram of the system is shown in Figure 4. The 12 V power supply is reduced to 3.3 V by two linear voltage-regulator chips, which are used to power the main control chip. The STM32 microcontroller communicates with the leaf expansion sensor probe, temperature and humidity sensor HDC1080, and light sensor LTR-303 through the IIC interface. Communication with the soil moisture sensor and the SD card is performed via the SPI interface. Data transmission with LoRa is performed via the URAT interface.

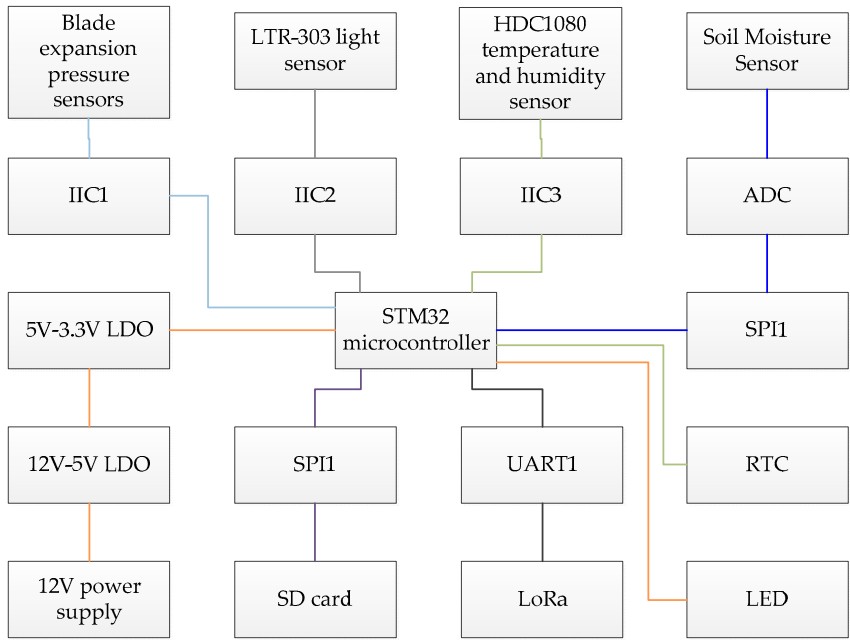

**Figure 4.** System simplified circuit diagram.

## 2.2.2. Power Supply Circuit Design

The power supply circuit module is the basic unit of the sensor, supplying power to each module and ensuring the effective operation of each module. Since the power supply is a 12 V lithium battery, the LoRa communication circuit requires a 5 V power supply, and the STM32 processor and core circuit, sensor module, and SD card all require a 3.3 V power supply; therefore, the input power supply needs to be stepped down. In Figure 5, CJ78M05 and RT9013-33GB linear regulators are used to step down the 12 V supply voltage to 5 V and 5 V to 3.3 V, respectively.

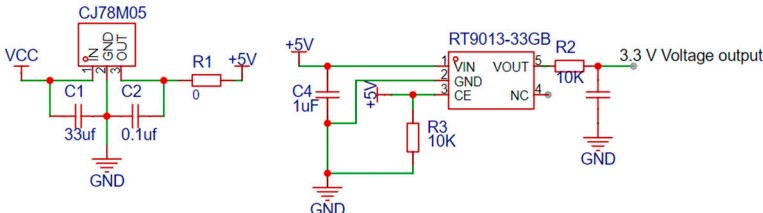

**Figure 5.** Schematic diagram of power supply circuit.

## 2.2.3. Sensor Monitoring Circuit Design

The system is also equipped with many sensors for data measurement and analysis. The circuit diagram is shown in Figure 6. The leaf expansion pressure detection unit uses a high-precision pressure probe to determine the expansion pressure of the leaf by sensing the pressure value of the leaf output at a constant pressure. The temperature and humidity detection unit adopts the HDC1080DMBR high-precision digital sensor; its temperature detection error is ±0.2 °C, and its relative humidity error is ±2%. The light intensity detection unit adopts an LTR-303ALS-01 high-precision digital sensor, which can achieve a wide range of dynamic light detection from 0.1 lx to 64,000 lx. The soil moisture detection unit uses a JXBS-3001-TR conductivity soil moisture sensor, which can achieve measurement accuracy within 3%. The temperature and humidity detection unit, and light intensity detection unit, are used to detect the temperature and humidity of the measured leaves under the same time space and light-intensity data; the soil moisture detection unit is used to detect the soil moisture of the measured plants; and the expansion pressure sensor data are used for comparison and analysis. The leaf expansion pressure and the light,

temperature, and humidity detection unit transmit digital signals to the MCU through the IIC communication protocol; the soil moisture detection unit outputs analog voltage values, which are converted to digital signals by the ADS1255 chip and transmitted to the MCU.

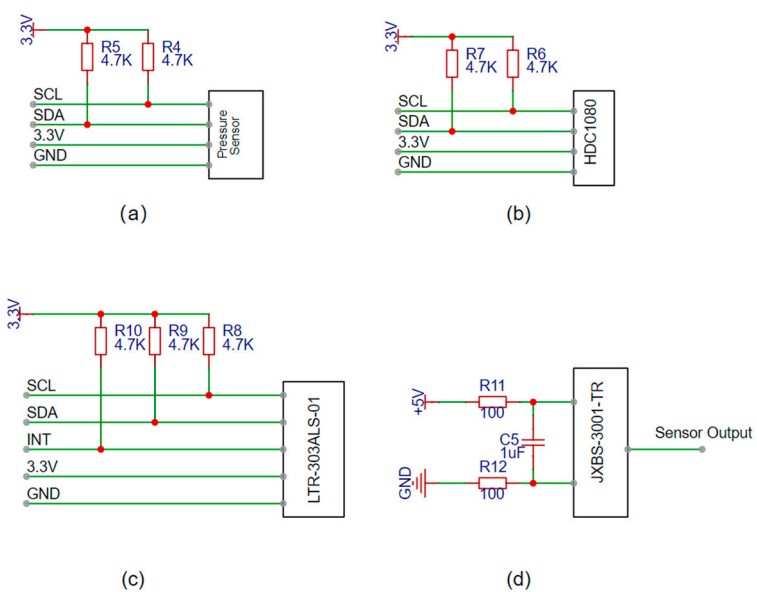

**Figure 6.** Sensor Interface Circuit: (**a**) Interface circuit of expansion probe. (**b**) Temperature and humidity monitoring module circuit. (**c**) Light-intensity monitoring module circuit. (**d**) Soil moisture sensor.

### 2.3. Sensor Experiments

#### 2.3.1. Sensor Calibration Experiments

To determine the pressure value corresponding to the digital quantity collected by the microcontroller, a standard weight was selected for its calibration. Standard weights are calibrated by high-precision balance. In the analysis of the experimental blade expansion pressure measurement data, the measurement range is 10–170 kPa at best. The pressure is converted to the data under the gravity of the probe, corresponding to the probe area under the gravity of a mass of 78–1334 g. Considering that the maximum value of the measurement should be less than 2/3 of the range, the calibration range is chosen as a 0–2000 g mass of gravity. In turn, the expansion pressure detection unit is placed on different masses of weights (10, 50, 100, 200, 500, 1000, 1500, and 2000 g) that record the sensor output data.

#### 2.3.2. Sensor Stability Test Experiment

In the case of the pressure probe and the magnet, without clamping the leaf between the probe and with the measurement pressure side up and horizontally fixed, continuous static measurements for 600 min every 10 min recorded the sensor data.

#### 2.3.3. Leaf Expansion Pressure Measurement Test

To verify the feasibility of the sensor, the overall experiment was divided into two phases and tested on the Green Rose (*Epipremnum aureum*) and the Umbellata (*Ficus umbellata*).

The first phase of the experiment was conducted from May to July 2022 in the laboratory of the engineering science building of Beijing Forestry University (40°00′08″ N; 116°20′59″ E; 50 m above sea level). The region has a warm, temperate, and semi-humid continental monsoon climate with cold and dry winters; hot and rainy summers; short springs; short autumns; variable spring temperatures; cool autumns; abundant light, rain, and heat in the same season; and significant seasonal changes. The annual average temperature is 12.5 °C, the extreme minimum temperature is −21.7 °C, and the extreme maximum temperature is 41.6 °C. The annual sunshine count is 2662 h, and the frost-free period is

211 days. After adequate irrigation of the Green Rose, the leaf expansion pressure sensor probe was mounted on the upright green, healthy leaves. The installation is shown in Figure 7 with the side with the pressure sensor placed on the back of the leaf while avoiding the leaf veins to prevent measurement errors caused by the veins and dust on the leaf surface.

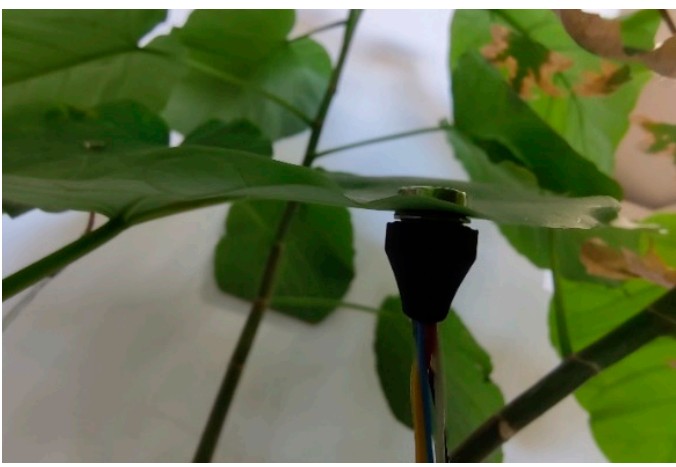

**Figure 7.** Measurement chart of swelling pressure probe.

Firstly, we tested the change of leaf expansion pressure under different moisture levels and subjected the *Epipremnum aureum* to high water stress treatment to observe the change of leaf expansion pressure from sufficient to high water stress. Secondly, under high water stress, they were irrigated adequately, and the changes in leaf expansion pressure received by the microcontroller were recorded. Then, we investigated the effect of environmental factors on leaf expansion pressure and recorded the temperature, humidity, light, and expansion pressure values received by the microcontroller simultaneously for analysis.

The second phase of the experiment was conducted from August to September 2022 in the outdoor corridor of the engineering building of Beijing Forestry University, which is directly connected to the outdoors through a window, to simulate the outdoor environment and to avoid uncontrolled moisture due to outdoor rain. The sensor probes were placed on the back of a slightly shrunken leaf and a healthy and upright leaf. The plant was then tested over time and irrigated midway to avoid excessive water depletion.

The blade pressure probe was measured as shown in Figure 7, and the sensor probe was attached to the blade by a magnet. The plant leaves used in this experiment were relatively upright. If a softer leaf is used for testing, the probe needs to be supported to avoid deformation of the leaf.

## 3. Results

### 3.1. Calibration Experiment Results

The calibration curve obtained by linearly fitting the data collected by the sensor to the mass of the weights is shown in Figure 8. There is a good linear relationship between the digital quantity and the pressure, and the coefficient of determination of the fitted equation is above 0.99, indicating that the detection unit can accurately measure the pressure value.

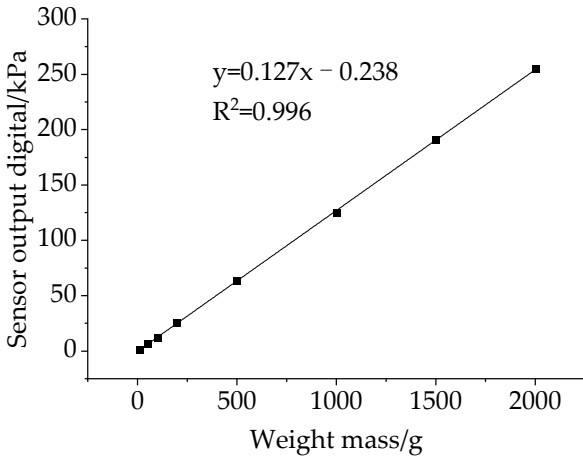

**Figure 8.** Calibration curve of expansion pressure detection unit.

### 3.2. Sensor Stability Test Results

The stability test of the sensor at room temperature (21 °C) is shown in Figure 9. The sensor output value is stable within 600 min continuously at room temperature, and the pressure fluctuation is within 0.2 kPa, which can meet the requirements of use.

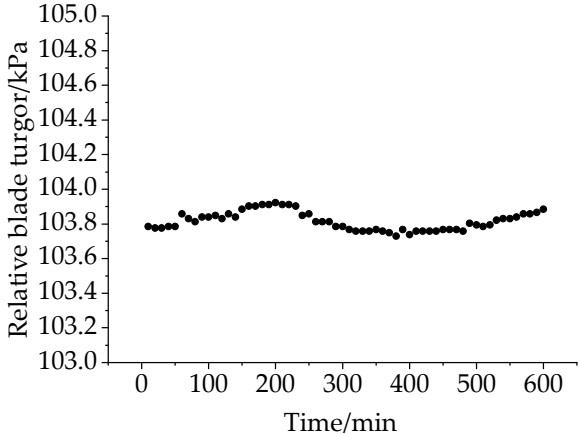

**Figure 9.** Sensor stability test results.

### 3.3. Leaf Expansion Pressure Measurement Test Results

3.3.1. Leaf Expansion Pressure under Different Moisture Conditions

The leaf expansion pressure measurements were carried out in a single day for the leaves under different moisture conditions. In the normal moisture state, the expansion pressure curve showed a "day-high and night-low" state, as shown in Figure 10. $P_p$ started to rise at about 7:00 a.m. and continued to increase until about 13:00 h, when it reached the peak, with a peak duration of about 2 h, and then gradually decreased.

After stopping irrigation for a period of time, it was observed that the leaves of the plants showed obvious shrinkage and water deficiency, during which the changes in leaf expansion pressure were continuously monitored. When the plant was under very dry conditions, the leaf expansion data curve showed the opposite trend to the normal condition, with an overall "day-low and night-high" shape. The data are shown in Figure 11.

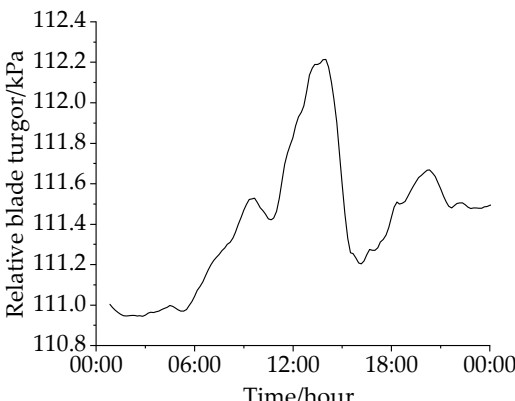

**Figure 10.** The turgor pressure curve under sufficient moisture.

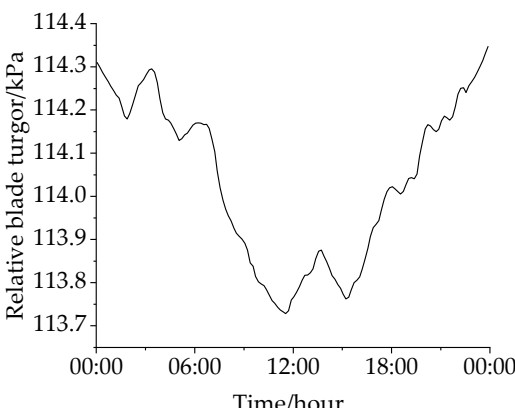

**Figure 11.** Turret pressure curve under high water stress.

At the moment of change from normal to water deficit, a "semi-inverted" curve was observed, which showed a rise in expansion pressure in the morning, reaching a peak, and then a rapid decline in expansion pressure data, which would drop to a minimum value at midday, as shown in Figure 12. Only afterward did it rise slowly, and the curve was inverted by half compared to the expansion pressure when water was sufficient.

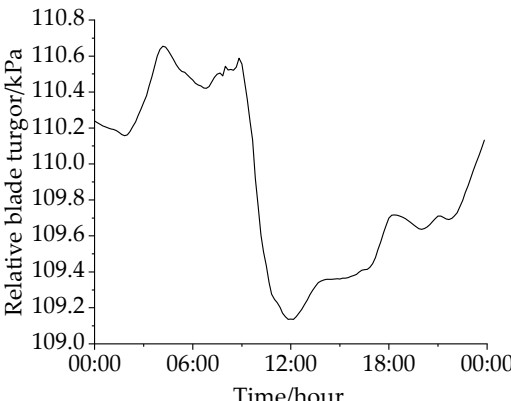

**Figure 12.** Turret pressure curve under preliminary water stress.

By analyzing the expansion pressure data of the three conditions, it was found that the leaf expansion pressure was higher than normal when the plant was in a high-water-deficit condition. However, the relative expansion pressure variation was also reduced. Theoretically, when an extreme drought causes a lack of water in the leaves, the ratio of water to air in the leaves is very poor, and $P_c$ is very small; this results in $F_a$ becoming

the main factor affecting the expansion pressure, resulting in data showing a "day-low night-high" phenomenon, and the overall value being greater than the normal state. The data are shown in Table 1.

**Table 1.** Change parameters and amplitude of leaf turgor pressure under different water conditions.

| Data Type | Minimum Value/kPa | Maximum Value/kPa | Relative Variation |
|---|---|---|---|
| Sufficient water | 110.9 ± 0.2 | 112.3 ± 0.2 | 1.0 |
| Initial water shortage | 109.1 ± 0.2 | 110.7 ± 0.2 | 1.1 |
| Severe water shortage | 113.6 ± 0.2 | 114.4 ± 0.2 | 0.5 |

### 3.3.2. Irrigation Expansion Pressure Change under High Water Shortage

When the plants were under an obvious water deficit, a sufficient amount of irrigation was applied; the relative leaf expansion pressure $P_p$ is shown in Figure 13, which shows a significant downward trend from the high value of the water deficit to a low value, and the data curve for the following days shows a normal "high-day and low-night" state.

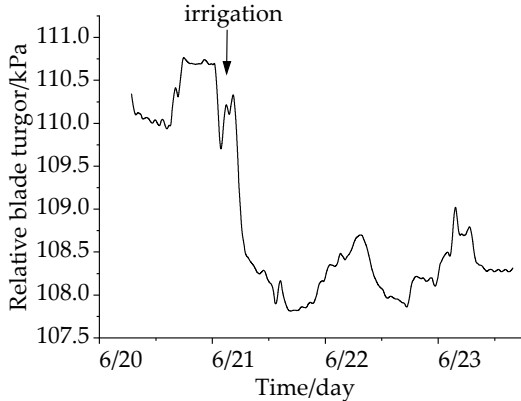

**Figure 13.** Change curve of swelling pressure after irrigation.

### 3.3.3. Correlation Analysis between Expansion Pressure and Environmental Factors

To study the correlation between plant leaf expansion pressure, ambient temperature and humidity, and light intensity, the data were divided into three groups: high-light (light intensity > 100 lx), low-light (light intensity < 100 lx), and all-day data, which were subjected to Pearson correlation analysis. The data showed that the correlation coefficient between expansion pressure and light intensity was the largest, and the correlation coefficient of all-day $r$ could reach 0.817, as shown in Table 2.

**Table 2.** Correlation between leaf turgor pressure and environmental factors.

| | High-Light Group | | Low-Light Group | | All-Day Group | |
|---|---|---|---|---|---|---|
| | $r$ [1] | $p$ [2] | $r$ | $p$ | $r$ | $p$ |
| Turgor pressure–temperature | −0.250 | 0.882 | −0.353 | 0.000 | 0.459 | 0.000 |
| Turgor pressure–humidity | 0.321 | 0.530 | −0.328 | 0.001 | −0.473 | 0.000 |
| Turgor pressure–Light intensity | 0.650 | 0.000 | 0.561 | 0.000 | 0.817 | 0.000 |

[1] Correlation coefficient between variables. [2] Significance coefficient.

### 3.3.4. Continuous Monitoring Data Analysis

In order to verify the long-term stability of the sensor, the second phase of the experiment was chosen to conduct long-term stationary measurements on the tested plants to explore the

changes in leaf expansion pressure over a longer period of time. There were two measurement points on the tested plants: measurement point one was located 1.6 m from the ground, and the tested leaves were mature, with an overall slightly shriveled water-deficit trait; measurement point two was located 1.2 m from the ground, and the leaves were new, healthy leaves. Both measurement points were not blocked by light, and light conditions were comparable. The shriveled leaf relative pressure data overall showed the "day-low night-high" phenomenon, and the healthy leaf relative pressure data overall showed the "day-high night-low" phenomenon. The two daily changes were close to the opposite, with the shriveled leaf relative pressure overall being higher than the healthy leaf. As seen in the fifth column of Figure 14, the data curves for healthy leaves (red line) and shriveled leaves (black line) show opposite trends. The peak of the healthy leaf expansion pressure wave is at the same moment as the trough of the shrunken leaf. It is obvious that the shrunken leaf data are larger than the healthy leaf.

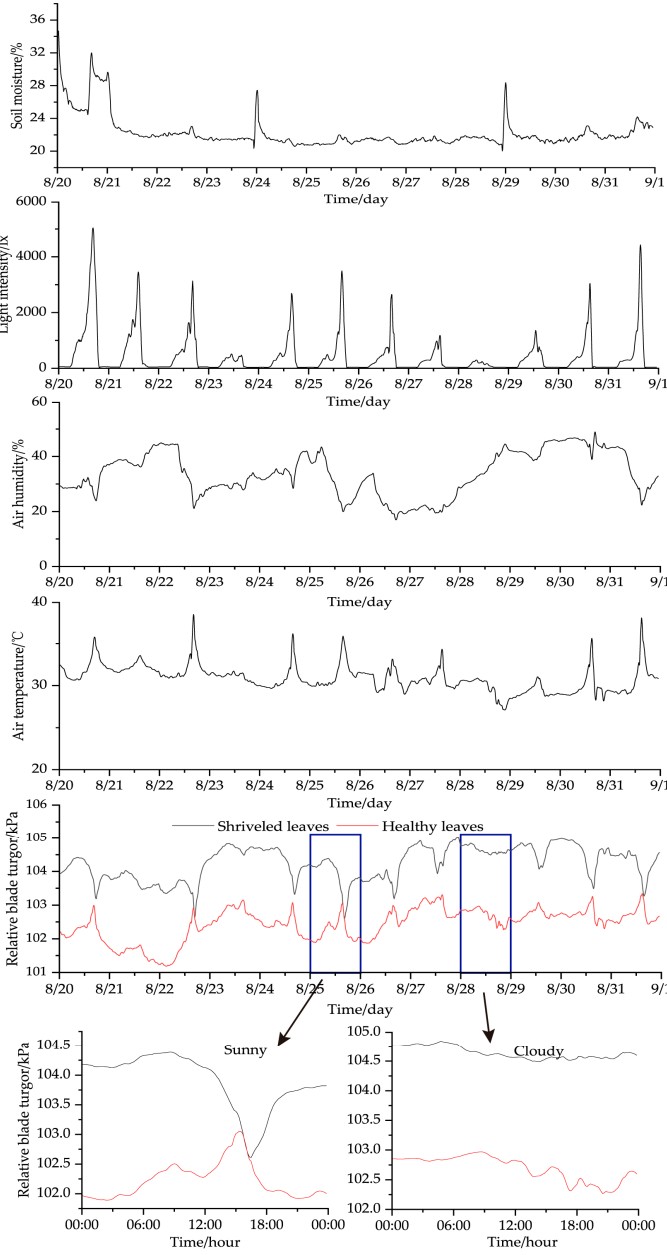

**Figure 14.** Changes in relative leaf turgor pressure under long-term monitoring and different weather conditions.

The relative expansion pressure in different weather varied, with the relative leaf expansion pressure in sunny weather increasing by 6.7% compared to the minimum value in 1 day, while in rainy weather, it only increased by 3%. This means that the value of the leaf expansion pressure changes little under rainy weather. This is partly due to the fact that the transpiration of the leaves is reduced in rainy weather, resulting in small changes in plant water. The curves are shown in Figure 14.

As can be seen in Figure 14, whenever irrigation is performed or during periods of cloudy weather with high air humidity, leaf expansion pressure values decrease to varying degrees relative to drought conditions. This indicates that leaf expansion pressure is a good indicator tool for plants under different water conditions and can be used to guide irrigation. However, this first requires collecting a large amount of data to determine the expansion pressure threshold when plants are water-deprived, which is the next step in our research.

## 4. Discussion

A non-destructive detection sensor for plant leaf expansion pressure based on LPCP technology was designed, which can realize long-term non-destructive online in situ measurement during plant growth and provide an effective technical means for plant leaf expansion pressure monitoring and research. The calibration experiment results of the device show that the designed sensor measurement results, and the true value of the fit coefficient of determination, were higher than 0.99. The static characteristics of the experiment show that the sensor stability is good, and the pressure fluctuation error is within 0.2 kPa. This indicates that the sensor can effectively monitor the changes in leaf expansion pressure and can be used as an effective monitoring tool for plant leaf expansion pressure research. By monitoring the expansion pressure changes in the artificially controlled environment of the indoor Green Rose, and the long-term expansion pressure changes of the outdoor Ficus umbellate, it can be seen that after stopping irrigation, the leaf expansion pressure changes in three curves as the water decreases; after irrigation, the overall value of leaf expansion pressure decreases. The correlation coefficient between leaf expansion pressure and light intensity was 0.817, and the long-term outdoor data showed that the relative leaf expansion pressure changed differently in different weather conditions. The change of expansion pressure in sunny weather was more significant than that in rainy weather. The study of the response of plant leaf expansion pressure to water indicators was expanded from intermittent indicators to continuous parameters. The relationship between multiple factors and leaf expansion pressure will be studied in depth. Based on this study, the quantitative relationship between plant leaf expansion pressure variation and water can be further derived to specify irrigation strategies and achieve precise irrigation.

**Author Contributions:** Conceptualization, Y.Z.; methodology, X.M. and Y.Z.; software, X.M.; validation, X.M.; investigation, X.M.; writing—original draft preparation, X.M. writing—review and editing, X.M. and Y.Z.; visualization, X.M.; supervision, Y.Z. and W.L.; project administration, Y.Z. and W.L.; funding acquisition, Y.Z. and W.L. All authors have read and agreed to the published version of the manuscript.

**Funding:** This research was funded by "The Fundamental Research Funds for the Central Universities", grant number 2021ZY74.

**Institutional Review Board Statement:** Not applicable.

**Informed Consent Statement:** Not applicable.

**Data Availability Statement:** The data used in this study are reported in the figures and tables of the paper.

**Conflicts of Interest:** The authors declare no conflict of interest.

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
