# Peer review of "Development of Non-Destructive Testing Device for Plant Leaf Expansion Monitoring"

_electronics, doi:10.3390/electronics12010249_

Round 1
Reviewer 1 Report
The topic of the article is relevant to applied Journal. The interdisciplinary research make the article interesting. Anyway better precision in describing the experiment, better quality of Figures and wider conclusions are required. In my opinion, the text could be accepted after a major revision.
The recommendations are as follows:
1. Figures:
a) Figures 1, 5-9 are poor quality with visible pixels. That should be improved and .tiif or other vector pictures could be used
b) Punctuation mistakes in description of Figure 2. – missing space between dot and description.
c) Figures 1,2, 3, 4, 9 were not mentioned and described in the text.
d) The fonts used in Figures differ and it is messy. It should be unified to the font demanded by Journal.
e) If the Figure 1 is copied from another literature, the citation should occur in the description.
f) In Figure 12 fluctuations of 0.2 kPa are poorly visible – some zoom on Y scale could be used to present the data better.
g) In Figure 3. what is ‘suppli’? Should it be ‘supply’?
h) Figures 6-9 could be presented in 2x2 order to reduce space.
i) The citations of figures in text should be unified; sometimes it is Figure, sometimes Fig (see line 282).
2. Abstract
a) It is kPa, not ‘kpa’ like in lines 18,19, 247.
b) In line 27 the expression ‘many days’ is used. If we use engineering language the period of this should be precisely expressed – how many days, hours etc.
3. When we used square [] brackets the space between text and bracket is necessary – see lines 39,42,47 and other
4. In Equation 1 the ‘*’ sign is used. If it corresponds to multiplying (not Convolution) the ‘×’ or ‘·’ should be used instead.
5. The Pclamp parameter is described with different font in text and equation – it should be unified.
6. Punctuation mistakes while describing equations –see in line 96 – additional space before ‘:’ and in line 98 ‘While’ should start with small letter ‘w’. Same in lines: 113,166.
7. Instead of ‘~’ sign the word ‘to’ or ‘-‘ sign should be used – lines 188, 190.
8. In line 188 the ‘range of 10kpa ~ 170kpa ranging’ should be rewritten properly.
9. In 196, 202 lines (and in whole text) the spaces between number and units (such as m, min, g etc.) should be used.
10. Lines 221-226 are multiplied in lines 227-232.
11. In line 254 the ‘Pp’ should be rewritten with indexing.
12. In Table 1, the results of pressure are given with 6 significant digits. The measurement precision cannot be higher than the level of fluctuations: 0.2 kPa. That lead to the conclusion that the number of digits should be limited to 4, and the measurement error ±0.2 kPa should be presented.
13. In the description of Table 1, what does it mean: ‘under different water’?
14. Authors mentioned that the cost of commercially available equipment is about 1500$, but did not presented the calculation of their design. Is the cost of the presented solution much lower? If yes, the comparison should be presented as an argument for being competitive.
15. In line 186 it was mentioned that the ’standard weight was selected for (..) calibration’ of the probe. What was this ‘standard mass’? Was it calibrated due to any international standard or probe specification? If yes, it should be cited. If not, the calibration should be described more precisely.
16. The difference between maximum and minimum values of pressure presented in Figures 13-1 5 is about 1.5 kPa. Due to fluctuations (0.2 kPa), the relative accuracy is quite high reaching 13%. If better accuracy is not possible to obtain due to some experimental limitations it should be widely described and discussed.
17. In line 290 the unit should be unified (lux or lx ?).
18. The p coefficient was not determined in the text (see Table 2).
19. Values of parameters in Table 2 are given with different number of significant numbers (sometimes 2, sometimes 3). It should be unified.
20. The data presented in Figure 17 was poorly described. Only in lines 307-310 I found some precise conclusions proved by numbers. Calculations presented in lines 307-310 and the results presented in Figure 17 should be widely discussed.
21. The conclusions given in lines 296-306 are not proved by any results – they are just an observations. Some argumentation (photos, measurements) should be given to prove this statement.
22. Air humidity fluctuations are about 25%. What is the cause of such huge fluctuations?
23. The literature should be unified due to Journal demanding.
Reviewer 2 Report
The results were very interesting and corroborated with the literature. However, the authors left something to be desired in the discussion of the results, they could have explored the discussion further, emphasizing the importance of this study;
Work the discussion further.
Reviewer 3 Report
Dear authors,
The topic of research is really interesting and could have a high impact on modern agriculture. However, you need to drastically improve the English language used. Sentences are long with syntax errors making it difficult for the reader to understand your text.
Abstract: Please rewrite, it is hard to understand. Too lengthy sentences and poor use of English.
Line 39: Use the extended version before using abbreviations, Soil Plant Atmosphere Continuum (SPAC)
Line 57: Remove the price in dollars
Line 60-61: Improve English, hard to read
Line 62: Replace LORA with LoRa and explains it means Long Range
Line 64-65: Rephrase it is hard to understand
Line 91-96: Hard to understand
Line 188: Rephrase hard to understand
Line 199: Please also provide the latin names
Line 259-260: Rephrase hard to understand
Author Response
I apologize for missing the attachment the first time, I resubmitted it again, I don't know if you received the attachment?

Round 2
Reviewer 1 Report
Authors responded to my review and they improved the text.
Some minor language mistakes still are present and it should be revised once again properly. Examples below:
- in line 334 'Figuer', it should be 'Figure'
- Figures 2, 3,4,7,8 still were not mentioned in the text and not commented
- in the description of Table 1, the expression 'under different water' should be explained the way it was written in the coverletter - ex. 'under different water conditions'
Author Response
Please see the attachment。

Reviewer 3 Report
Dear authors,
Thank you for addressing all comments.
Author Response
Thank you for your review of this article.